# Calcium signals are necessary to establish auxin transporter polarity in a plant stem cell niche

Ting Li [1,2], An Yan[1,2], Neha Bhatia [3], Alphan Altinok[4], Eldad Afik[1], Pauline Durand-Smet[1], Paul T. Tarr[1,2], Julian I. Schroeder[5], Marcus G. Heisler [3] & Elliot M. Meyerowitz[1,2]

In plants mechanical signals pattern morphogenesis through the polar transport of the hormone auxin and through regulation of interphase microtubule (MT) orientation. To date, the mechanisms by which such signals induce changes in cell polarity remain unknown. Through a combination of time-lapse imaging, and chemical and mechanical perturbations, we show that mechanical stimulation of the SAM causes transient changes in cytoplasmic calcium ion concentration ($Ca^{2+}$) and that transient $Ca^{2+}$ response is required for downstream changes in PIN-FORMED 1 (PIN1) polarity. We also find that dynamic changes in $Ca^{2+}$ occur during development of the SAM and this $Ca^{2+}$ response is required for changes in PIN1 polarity, though not sufficient. In contrast, we find that $Ca^{2+}$ is not necessary for the response of MTs to mechanical perturbations revealing that $Ca^{2+}$ specifically acts downstream of mechanics to regulate PIN1 polarity response.

[1] Division of Biology and Biological Engineering, California Institute of Technology, 1200 East California Boulevard, Pasadena, CA 91125, USA. [2] Howard Hughes Medical Institute, California Institute of Technology, 1200 East California Boulevard, Pasadena, CA 91125, USA. [3] School of Life and Environmental Sciences, The University of Sydney, Darlington, NSW 2008, Australia. [4] Machine Learning and Instrument Autonomy, NASA Jet Propulsion Laboratory, California Institute of Technology, 4800 Oak Grove Drive, Pasadena, CA 91109, USA. [5] Division of Biological Sciences, Cell and Developmental Biology Section, University of California, San Diego, La Jolla, CA 92093, USA. The authors contributed equally: Ting Li, An Yan, Neha Bhatia. Correspondence and requests for materials should be addressed to M.G.H. (email: marcus.heisler@sydney.edu.au) or to E.M.M. (email: meyerow@caltech.edu)

Plant cells respond in a number of ways to mechanical stresses. Cortical microtubules (MTs) align to the direction of maximal anisotropic mechanical stress in shoot apical meristems (SAMs) and in pavement cells[1,2]. As MTs guide cellulose synthesis[3], this is thought to lead to directional reinforcement of the cell wall[1,2]. Polar transport of the plant hormone auxin is necessary for floral organ initiation and tissue morphogenesis in SAMs[4]. In the epidermal cells of the SAM, the plasma membrane-localized transporter of auxin PIN-FORMED 1 (PIN1) is the major auxin efflux carrier that directs auxin flow, and as a consequence controls the formation of leaves and flowers[5–7]. The PIN1 protein exhibits dynamic patterns of expression and polarity and its loss of function is sufficient to disrupt floral phyllotaxis[8]. It has been proposed that the asymmetric distribution of PIN1 in the plasma membrane of inflorescence meristem epidermal cells (PIN1 polarity) is also patterned by mechanical signals, as its membrane localization responds to mechanical changes and correlates with the orientation of cortical MTs[1,6]. The current model proposes that PIN1 adjusts its subcellular location to be preferentially in the plasma membrane adjacent to the most stressed side wall[6], which directs auxin toward expanding neighbors (as auxin causes wall weakening and, therefore, cell expansion). This positive feedback controls auxin flow in the epidermis, and as a consequence controls the pattern of formation of leaves and flowers[6,7]. The mechanisms by which mechanical signals induce changes in meristem cell polarity remain to be determined.

An additional cellular response to mechanical stress is mechanically induced transient change in cytoplasmic calcium ion concentration ($Ca^{2+}$)[9,10]. It is known that $Ca^{2+}$ signaling plays a role in the development of patterning and morphogenesis in early embryos in ascidians, frogs, and zebrafish. In plants, de la Fuente et al. studied the role of $Ca^{2+}$ in auxin transport[11]. Auxin application causes a rapid increase in cytosolic $Ca^{2+}$ in *Zea mays* coleoptiles and plant protoplasts[12–14], and $Ca^{2+}$/calmodulin binding to auxin-induced regulatory proteins has been suggested to control auxin response in maize roots[15]. These and many other studies indicate that calcium-mediated processes may be involved in modulating auxin response and polar auxin transport[16,17]. Yet the link underlying $Ca^{2+}$ signaling and morphogenesis is not fully understood. Here we show that one prerequisite for the PIN1 response, which takes place over hours, is the much more rapid calcium response, as blocking the calcium response prevents later PIN1 relocalization. This is not a result of irreversible injury to the meristem cells and this is necessary for initiation of PIN1 relocalization and not for the protein vesicle traffic itself. The findings in this study establish a new framework for the function of $Ca^{2+}$ signals in cellular polarity during organ initiation, tying them directly to morphogenesis.

## Results

### Blocking $Ca^{2+}$ signals prevents PIN1 repolarization in growth.

To explore the possibility that $[Ca^{2+}]_{cyt}$ signals are related to PIN1 protein dynamic changes during tissue growth, we immersed dissected *pPIN1::PIN1-GFP* transgenic inflorescence SAMs in $LaCl_3$ (5 mM), known as a plasma membrane $Ca^{2+}$ channel blocker[18], for 2 min every hour for 12 h without rinse after each treatment to maintain the chemical effect. Control samples were treated identically, using water without the inhibitor. After mock treatment, SAMs continued growing (cell division can be detected in every sample, Supplementary Fig. 1a), producing new floral primordia (Fig. 1a, c) and showing stereotypical changes in PIN1-GFP subcellular polarization[7] ($n = 10$) (Fig. 1a–d). $LaCl_3$-treated meristems formed no new flower primordia, and the PIN1-GFP polarization pattern remained

unchanged, with a gradual increase in overall fluorescent signal intensity ($n = 10$) (Fig. 1e, f, h, i). Cell division was seen in every $LaCl_3$-treated sample (Supplementary Fig. 1a). The $LaCl_3$ blockage effect diminished after a 12 h recovery in which the samples were rinsed in water. After this recovery PIN1-GFP regained the ability to repolarize (Fig. 1g, j). To control for any non-specific effects of $LaCl_3$, we used a second and unrelated method to block calcium response, treating meristems with 1,2-bis(o-aminophenoxy)ethane-N,N,N′,N′-tetraacetic acid (BAPTA), an extracellular pH-insensitive calcium chelating agent[19,20], and tested its effect on PIN1 protein polarization dynamics during growth. We found that BAPTA also caused a delay or inhibition of PIN1-GFP polarity convergence on new primordia, although in this case no PIN1-GFP fluorescent signal increase was detected ($n = 10$ for BAPTA-treated SAMs, $n = 6$ for mock-treated SAMs, Supplementary Fig. 2a–j). Cell divisions still occurred during BAPTA treatment, but the frequencies were lower compared to mock-treated samples (Supplementary Fig. 1b).

### $Ca^{2+}$ signal blockage prevents PIN1 mechanical repolarization.

To see if inhibition of calcium response affects the PIN1 repolarization that results from external mechanical cues, we used local cell ablation treatments that change the mechanical stress pattern in the SAM, normally causing a predictable pattern of PIN1 relocalization over several hours[6]. We pretreated dissected *pPIN1::PIN1-GFP* transgenic inflorescence SAMs with $LaCl_3$, then used a glass needle to kill a local group of cells. PIN1 response was monitored 3 h later by laser scanning confocal microscopy, as in Heisler et al. 2010[6] ($n = 5$) (Fig. 2a, b). In the meristems pretreated with 5 mM $LaCl_3$, no PIN1 relocalization response was observed in cells surrounding the ablation site ($n = 9$) (Fig. 2c, d). Partial inhibition of the relocalization response could also be observed with a pretreatment of 1 mM $LaCl_3$ when compared with mock treatment (Supplementary Fig. 3a–d, $n = 3$), but 5 mM $LaCl_3$ completely prevented PIN1 repolarization. After a 3 h recovery period during which the samples were rinsed in water without $LaCl_3$, PIN1 response to mechanical force change was restored, as PIN1-GFP relocalized around the earlier ablation site, similar to the control meristems (Fig. 2e, Supplementary Fig. 3e).

Similar effects to those with $LaCl_3$ were seen when the SAM was pre-treated with 2 mM BAPTA ($n = 14$ of 26 total treated meristems) (Supplementary Fig. 3f–i). The remaining 12 out of 26 BAPTA-treated samples showed only a weak PIN1-GFP signal suggesting that BAPTA apparently had a variable effect on PIN1 protein stability or degradation. Nevertheless, BAPTA treatment invariably caused inhibition of the PIN1 re-localization response in the cells where PIN1-GFP signal was visible. Restoration of extracellular calcium levels by rinsing samples with 3 mM $CaCl_2$ restored the normal PIN1 re-localization response (Supplementary Fig. 3j).

### $Ca^{2+}$ changes are not required for MT mechanical response.

A different mechanical response to that of PIN1 in the SAM cells is reorientation of the cortical MT cytoskeleton to the principal direction of maximal anisotropic stress in the cell wall[1]. In order to test the requirement for calcium during MT reorientation in response to mechanical perturbation, we carried out cell ablations using a multiphoton laser in inflorescence meristems pretreated with 5 mM $LaCl_3$ expressing both a *pML1::mCHERRY-MAP4* marker[21] along with *pPIN1::PIN1-GFP* as a positive control for $LaCl_3$ treatment. In contrast to PIN1-GFP, which did not reorient away from laser-ablated cells in the presence of 5 mM $LaCl_3$, we observed that MTs re-oriented circumferentially around the ablated cells despite $LaCl_3$ pre-treatment ($n = 5$ out of 6)

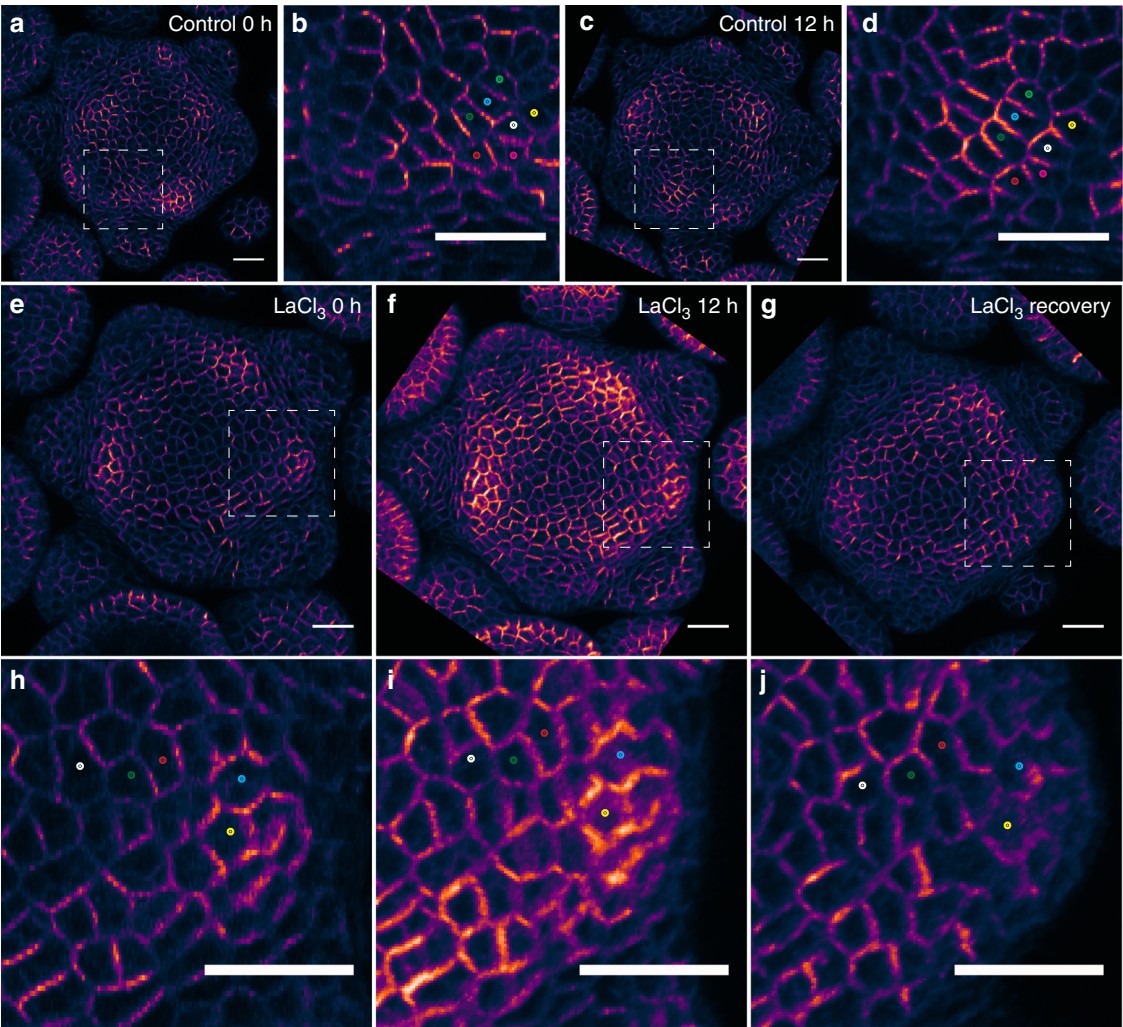

**Fig. 1** PIN1 dynamic polarization pattern during 12 h growth after 5 mM LaCl$_3$ treatment. **a**, **c** PIN1-GFP signal indicates a new primordium forming after 12 h (dashed square). **b**, **d** Enlargement of the square region in **a**, **c**. Colored dots indicate the cells that had PIN1-GFP polarity changed over 12 h of growth. **e–g** PIN1-GFP dynamic polarization pattern freezes after repeated application of 5 mM LaCl$_3$ for 2 min in every hour for 12 h without sample rinse (**f**), but primordia resume growth and PIN1-GFP signal relocalizes in the dashed square region after 12 h of recovery following water rinse (**g**). **h-j** Enlargement of the square region in **e–g**. The cells that are labeled by colored dots show that PIN1-GFP does not repolarizes during LaCl$_3$ treatment but resumes a dynamic pattern after a 12 h recovery. **a-j** scale bar: 20 μm. LUT is gem in Fiji-ImageJ software

(Fig. 2f–m, Supplementary Fig. 4). A similar MT response was observed after mechanical perturbation via glass-needle-induced cell ablation using a MBD-GFP MT marker after a pretreatment with 5 mM LaCl$_3$ or 2 mM BAPTA ($n = 5$ for LaCl$_3$ and $n = 6$ for BAPTA) (Supplementary Fig. 3k–n).

**Calcium dynamics in the SAM**. Having determined that calcium signaling is required for the reorientation of intracellular PIN1 polarity during regular growth and in response to mechanical perturbation, we next assessed [Ca$^{2+}$]$_{cyt}$ dynamics in the SAM by utilizing two fluorescent reporters of cytoplasmic Ca$^{2+}$ concentration, R-GECO1[22,23], and GCaMP6f(fast)[24]. We found that an individual SAM shows two different patterns of spontaneous Ca$^{2+}$ concentration dynamics, as inferred from fluorescence intensity. These patterns occur simultaneously and are similar to some aspects of calcium dynamics observed in animals, for example as characterized in larval organs in *Drosophila*[25]. Although both patterns consist of a transient increase in Ca$^{2+}$ concentration, one appears to be a global meristematic behavior, referred to as oscillations hereafter (Fig. 3a), while the other is at a

single cell level—sporadic single-cell transients, or calcium spikes as in the literature[26], referred to as spikes hereafter (Fig. 3b).

Analysis of plants expressing R-GECO1 indicates that the oscillations have a typical time between peaks of 22 min ± 4% (estimated mean ± relative standard error, standard deviation (SD) of 8 min 35 s, 123 inter-peak times), with event duration of 4 min 51 s ± 3%, estimate as full width at half maximum (FWHM) amplitude (mean ± relative standard error, SD of 1 min 53 s, $n = 141$ peak events). Supplementary Movie 1 shows one example of a SAM exhibiting the oscillations. The inter-peak time and FWHM are introduced graphically in Fig. 3a, further statistics are provided in Supplementary Fig. 5a, b and its caption; these findings are based on 19 SAMs out of 24, as explained in the SI section. Our observations do not indicate a single location in the SAM from which the oscillations originate, such as a new primordium initiation region, or any other meristematic landmark. Supplementary Movie 2 shows that over 5 h, oscillations initiated from a variety of SAM and primordia regions including central SAM region. To test whether the oscillations result from excision of the meristem from the plant, we imaged SAMs in intact plants, where similarly to the case of

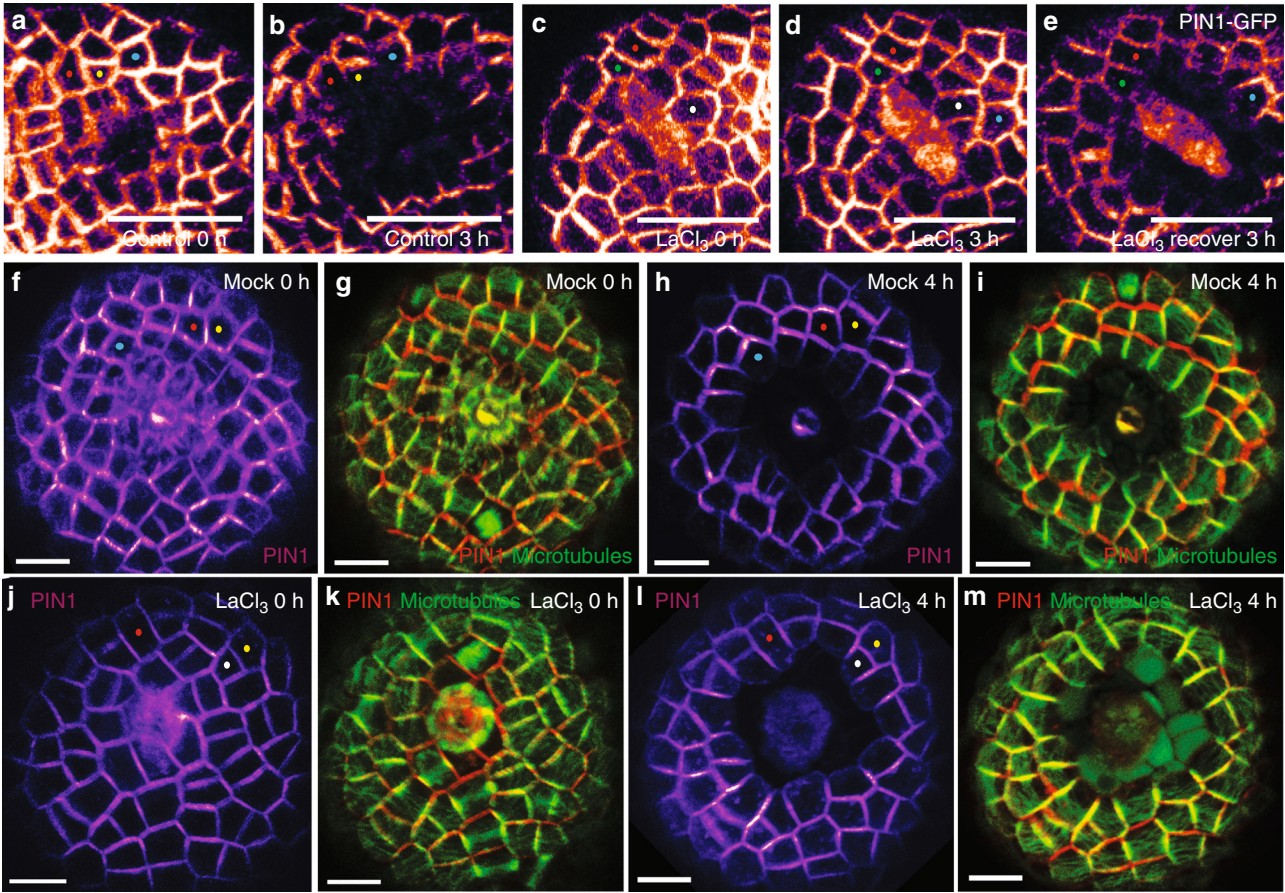

**Fig. 2** Mechanical responses of PIN1 and MT after LaCl₃ treatments. **a**, **b** PIN1-GFP shows polarized subcellular localization 3 h after local cell ablation with a glass pipette, without pharmacological treatment. **c**, **d** PIN1-GFP reorientation was not detected 3 h after cell ablation following 15 min pretreatment with 5 mM LaCl₃ without a subsequent water rinse. **e** At three additional hours after LaCl₃ washout, PIN1-GFP has reoriented away from the ablation site. **a**–**e** LUT is gem in Fiji-ImageJ software. **f**–**m** Confocal projections of laser-ablated SAMs expressing pPIN1::PIN1-GFP (magenta in **f**, **h**, **j**, **l** or red in **g**, **i**, **k**, **m**) and pML1::mcherry-MAP4 (green). **f**, **g** Mock treated meristem just after ablation. **h**, **i** The same meristem as in **f** and **g**, 4 h later. **j**–**m** Another laser ablated meristem, treated with 5 mM LaCl₃ 15 min before ablation and imaged at 0 h after ablation (**j**, **k**) and 4 h later without a water rinse during incubation (**l**, **m**). Note difference in PIN1 polarity response comparing **h** with **l** although MTs re-oriented circumferentially in both cases (compare **i** with **m**). **a**–**m** Similarly colored dots mark the same cells tracked over time. Scale bars: 20 μm (**a**–**e**) and 10 μm (**f**–**m**)

the excised SAM, hindering floral primordia and flowers had to be removed to allow imaging of the SAM tissue. In the intact SAMs, we found a typical time between peaks of 21 min ± 11% (estimated mean ± relative standard error, SD of 8 min 55 s, 15 inter-peak times), with event duration of 5 min 32 s ± 9% (FWHM, mean ± relative standard error, SD of 2 min 15 s, $n = 19$ peak events). Supplementary Movie 3 presents one example of an intact SAM; this is one out of the four SAMs included in the statistical analysis summarized above and in Supplementary Figure 5c and d; $n = 4$ out of 5 SAMs. The empirical distributions for the intact and excised SAMs, both for inter-peak-times and FWHM, are found to be so similar to each other that the null hypothesis that they are the same could not be rejected by the statistical tests we have applied (the smallest $p$-value was 0.28); see "Image processing and data analysis" under Supplementary Methods.

As for the single-cell spikes, these generally lasted between 1 and 5 s (more than 70% of events), though 13% of the cells showed prolonged signals lasting longer than 10 s ($n = 292$ individual cells from 30 SAMs in total events) (Supplementary Fig. 5f). A few cells appeared to spike repeatedly and continuously over minutes. Our observations indicate that one cell in hundred spikes every 20–40 s. Twelve SAMs showed only one spike signal

in more than 3 min in a normalized group of about 100 cells ($n = 69$ SAMs in the total experiments) (Supplementary Fig. 5e). Similar spikes and oscillations were observed in the cells of the meristems expressing the GCaMP6f sensor (Supplementary Movie 4), although at a higher frequency for the spikes, possibly due to higher sensitivity of the GCaMP6f sensor line. Both types of spontaneous Ca²⁺ dynamics were blocked when samples were immersed in 0.5 mM LaCl₃ ($n = 5$, Supplementary Movie 5) or in 0.2 mM BAPTA during time lapse imaging ($n = 5$, Supplementary Movie 6), which indicates that the spontaneous $[Ca^{2+}]_{cyt}$ transients are largely dependent on Ca²⁺ entry from apoplastic stores.

We next extended our observations to situations where the SAM is deliberately mechanically stimulated. After the meristems expressing the R-GECO1 sensor were mechanically stimulated by pressing from the side with a glass pipette (often accompanied by some cell injury at the point of contact) during confocal microscope observation (Supplementary Fig. 6a), a R-GECO1 fluorescent signal occurred as an intercellular calcium wave that initiated from the stimulated area (Fig. 3c, Supplementary Movie 7). After the calcium wave propagated through the SAM (Fig. 3d and Supplementary Fig. 7f), the Ca²⁺ signal started to decrease. Following the initial decrease, a dark crescent

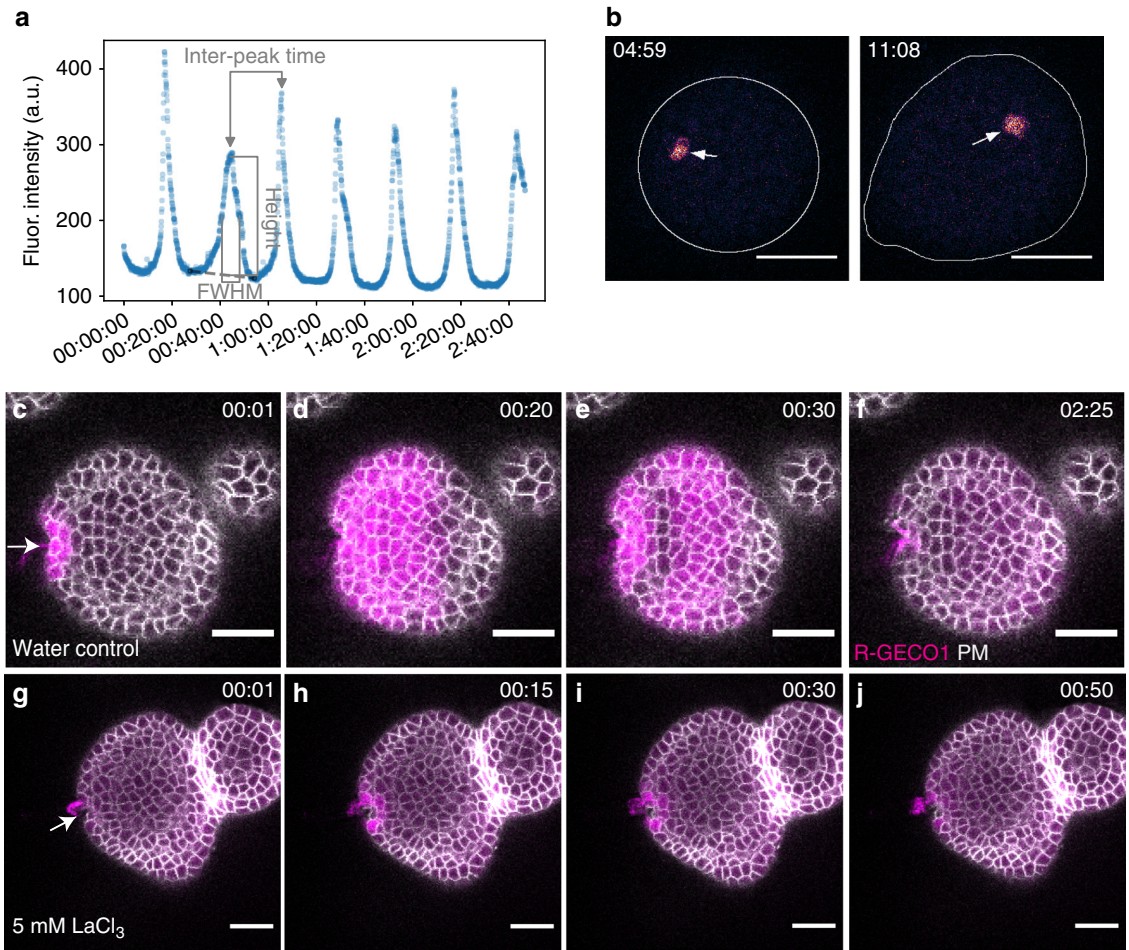

**Fig. 3** Three different types of $Ca^{2+}$ signal in the *Arabidopsis* SAM. $Ca^{2+}$ signals in the SAM include two spontaneous $Ca^{2+}$ patterns (**a**, **b**), and calcium waves that are activated by mechanical stimuli (**c–f**). **a** Global meristematic transient $Ca^{2+}$ levels, as measured by the mean fluorescence intensity level taken over a region-of-interest in the SAM, and plotted as function of time, showing regular peaks. The inter-peak time and the full width at half maximum (FWHM) are annotated in the panel, as well as the baseline (dashed gray line) and peak height. **b** Two representative $Ca^{2+}$ spike images. Arrow points to a cell showing a spike signal. The contour line delineates the border of the SAM. LUT is gem in Fiji-ImageJ software. **c–f** Representative frame images of R-GECO1 (magenta) and membrane marker 29–1 fused to GFP (gray) in the SAM, in response to mechanical stimulus from a glass pipette at the side of the SAM. **g–j** Effects of a 15-min pretreatment with 5 mM $LaCl_3$ on $Ca^{2+}$ wave transmission in the SAM. Arrow points to the direction of pipette prodding. Scale bars: 20 μm (**b-j**)

representing low $Ca^{2+}$ levels appeared between the stimulation site and the wave front at an average time of 23 ± 3 s (mean ± SD) and lasting for 16 ± 5 s (mean ± SD) (Fig. 3e, Supplementary Table 1). After 73 ± 18 s (mean ± SD) the calcium response was greatly diminished across the meristem, except in the cells closest to the stimulation site, which were the last to show decreased fluorescence ($n = 34$) (Fig. 3f). The $Ca^{2+}$ fluorescence increased nearly four-fold on average when compared with resting fluorescence levels (Supplementary Fig. 7e). We did a dilution series of $LaCl_3$ and BAPTA concentrations to observe the effects of different concentrations on mechanically induced $Ca^{2+}$ signals. On average $LaCl_3$ started to block the $Ca^{2+}$ wave propagation at 1 mM concentration (Supplementary Fig. 8a), and for BAPTA, 1 mM concentration is also capable of partially blocking the $Ca^{2+}$ waves (Supplementary Fig. 8b). In 5 mM $LaCl_3-$ and 2 mM BAPTA-pretreated SAMs, mechanical stimulation with a glass pipette did not induce calcium waves ($n = 26$ for $LaCl_3$, $n = 5$ for BAPTA) (Fig. 3g–j and Supplementary Fig. 7a–d, j), indicating that calcium wave initiation or propagation depends on plasma membrane $Ca^{2+}$ influx from apoplastic stores. Image analysis (Supplementary Fig. 9a–h) showed the average velocity of the

calcium wave signal transmission in the first half (by time) of the wave propagation to be about 2 μm s$^{-1}$, roughly one cell diameter per 2 s (Supplementary Fig. 7g). As the response proceeds across the meristem the speed slows down to about 0.5 μm s$^{-1}$ ($n = 22$ from 13 SAMs) (Supplementary Fig. 7g). The relatively low wave front speed indicates that the response fits into the slow calcium wave class[27]. In 5 mM $LaCl_3-$ and 2 mM BAPTA-pretreated SAMs, calcium wave velocities are close to zero ($n = 17$ from 17 SAMs for $LaCl_3$ and $n = 7$ from 5 SAMs for BAPTA) (Supplementary Fig. 7h, i). The GCaMP6f sensor showed a similar wave response to that of R-GECO1 upon laser-induced cell ablation (Supplementary Movie 8).

Next, we directly compared the spontaneous $Ca^{2+}$ oscillation signals and $Ca^{2+}$ wave following mechanical stimulation by a glass pipette using the same sample under the same imaging settings. We found that the spontaneous oscillation signals took 8.8 ± 4.4 times the interval of the time for mechanically triggered waves (mean ± SD, $n = 9$ individual SAMs). Thus the speed of spontaneous $Ca^{2+}$ signal propagation is much slower than that of deliberately mechanically stimulated waves. We also compared the fluorescence intensity changes between the two responses.

The maximum intensity fold change $(I - I_0)/I_0$ of the mechanically induced $Ca^{2+}$ wave is about $2.3 \pm 1.1$ (mean $\pm$ SD, $n = 7$); that of oscillation signals is $0.5 \pm 0.2$ (mean $\pm$ SD, $n = 12$) (Supplementary Fig. 10). The duration of the major peak of the wave is in about 2 min, but the oscillation signal duration time is longer than 10 min on average (Supplementary Fig. 10). Therefore, the endogenous oscillation reaction is weaker than the response to exogenous mechanical stimulation.

As prodding with a glass pipette appears to wound the SAM, we tested if non-injurious treatments had the same effect. As one example, in addition to responding when indented by a pipette, the SAM expressing the R-GECO1 sensor generates a comparable $Ca^{2+}$ wave when a glass pipette, pressed against the meristem, is withdrawn from contact (Supplementary Fig. 11a–d, Supplementary Movie 9). The response timeline is similar to that of the $Ca^{2+}$ wave propagated by the initial pipette stimulation (Supplementary Fig. 11i–k, Supplementary Table 1).

To further address whether the SAM $Ca^{2+}$ response can be triggered by noninvasive mechanical stimuli, we built a device that is able to press a coverslip down onto a cultured SAM from the top (Supplementary Fig. 6b, Supplementary Movie 10). After applying force with the coverslip, the R-GECO1 fluorescent signal did not increase compared with the resting level (Supplementary Fig. 11e, f). However, $Ca^{2+}$ fluorescence increased immediately upon SAM release from the cover slip (Supplementary Fig. 11g, l), with signal intensity change up to three-fold and the response lasting $22 \pm 7$ s (mean $\pm$ SD, $n = 14$) (Supplementary Fig. 11h, l). Previous studies have shown that compression stress is expected to reduce the stress intensity of a thin pressurized shell such that turgor pressure is locally balanced in the flattened area[28,29]. Conversely, when the planar constraint is released from the SAM, the previous high stress is expected to recur. Thus, it appears that intracellular $Ca^{2+}$ is more responsive to mechanical stress increase than decrease.

As mentioned above, the dynamic re-localization of PIN1-GFP is restored after diluting away $LaCl_3$ or BAPTA (Fig. 1g, j, Fig. 2e, Supplementary Fig. 3e, j). To see if the calcium response is similarly reversible, R-GECO1 reporter signals were observed in SAMs after reversal of earlier chemical treatments. Similar to the PIN1 response, $Ca^{2+}$ response returns after SAMs pretreated with either $LaCl_3$ or BAPTA were washed with water (Supplementary Fig. 12a–j, Supplementary Movies 11 and 12). In SAMs pretreated for 15 min with 5 mM $LaCl_3$, and incubated for 3 h, followed by a water washout, a $[Ca^{2+}]_{cyt}$ signal occurred after $8 \pm 3$ s (mean $\pm$ SD) and lasted for $23 \pm 8$ s (mean $\pm$ SD) (Supplementary Table 2). The primary peak amplitude of the whole-meristem fluorescent intensity was approximately two times that of the resting level. This was followed by a secondary peak that lasted for $81 \pm 12$ s (mean $\pm$ SD) with a mean amplitude of ~1.5 after an interval of $60 \pm 25$ s (mean $\pm$ SD) ($n = 11$) (Supplementary Fig. 12k, Supplementary Table 2). BAPTA-pretreated and then washed SAMs showed a similar response pattern ($n = 10$) (Supplementary Fig. 12l, Supplementary Table 2). As these $Ca^{2+}$ transients showed a biphasic kinetic pattern, and some of the transients were initiated away from ablated regions (Supplementary Movies 11 and 12), we suspect that these increases are likely to be caused by the washing treatment rather than mechanical stimulus.

**$Ca^{2+}$ response is required for PIN1 relocalization initiation.** To test whether $Ca^{2+}$ signals are necessary for PIN1 protein movement as it re-localizes (for example by being required for endocytotic or exocytotic vesicle traffic) or whether it may only be necessary for the initiation of the PIN1 response, we did meristematic cell ablation with a glass needle, waited for 5 min and

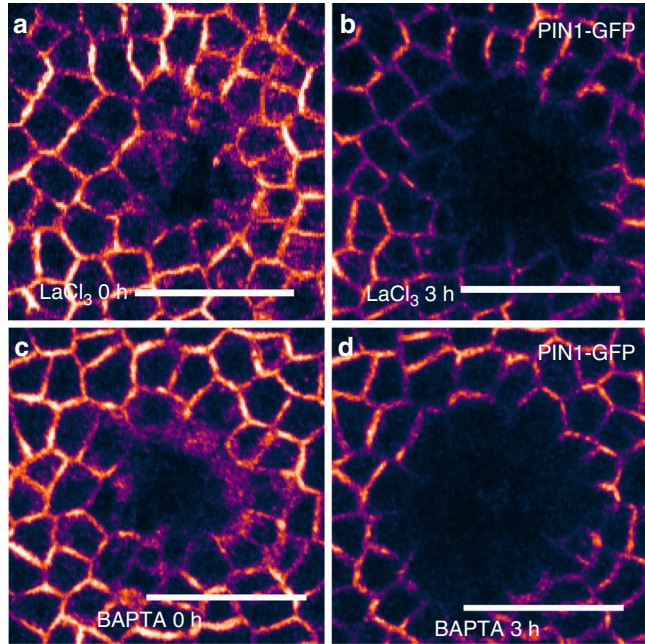

**Fig. 4** The PIN1 mechanical response is initiated within 5 min of stimulus and is not reversed by subsequent $Ca^{2+}$ inhibitor treatments. **a, b** PIN1-GFP behaves similarly to an untreated control at 3 h when pipette-mediated cell ablation precedes (by 5 min) the application of 5 mM $LaCl_3$ (which lasts 15 min with no subsequent washout). **c, d** PIN1-GFP behaves similarly to an untreated control at 3 h when pipette-mediated cell ablation precedes (by 5 min) the application of 2 mM BAPTA (which lasts 10 min with no subsequent washout), as in **a, b**. **a**–**d**, Scale bar: 20 μm. LUT is gem in Fiji-ImageJ software

then treated with 5 mM $LaCl_3$ or 2 mM BAPTA for 15 or 10 min, respectively, without a subsequent sample rinse, as in the earlier experiments. Three hours after stimulation, PIN1 had relocalized in both cases, just as if the SAM had never been treated ($n = 4$ for $LaCl_3$ and $n = 6$ for BAPTA) (Fig. 4). The results indicate that after mechanical stimulation, calcium response is only necessary for the initiation of the PIN1 response and is not required for later PIN1 protein trafficking.

## Discussion

The experiments described here provide a contribution towards understanding of how mechanical forces cause changes in the subcellular location of PIN1 protein, demonstrating that the initiation of PIN1 re-localization, and correspondingly, the initiation and the growth of floral primordia (but not reorganization of the cytoskeleton) require earlier changes in cytoplasmic calcium concentration. A previous report, on *Arabidopsis* roots, also demonstrated that changing $Ca^{2+}$ levels affects PIN protein localizations[30]. Recently, it has been reported that $Ca^{2+}$ transients participate in the AUX1-mediated root hair auxin influx and/or $SCF^{TIR1/AFB}$-based auxin signaling[16,17]. $Ca^{2+}$ signals thus play multiple roles in regulating tissue morphogenesis, affecting both auxin efflux and auxin signaling.

The mechanism of PIN1 relocalization is not known, but it involves redistribution of PIN1 protein from one membrane region to another by exocytosis and endocytosis that is partly determined by mechanical stress[31]. The major question, then, is how mechanical stress in cell walls translates to changes in the local balance of exocytosis and endocytosis of PIN1 transporting vesicles. What detailed mechanisms might connect calcium response to PIN1 relocalization? The normal asymmetric location of PIN1 in plasma membranes of multiple cell types, including

those of the SAM, requires activity of the PINOID gene, coding for a protein kinase that phosphorylates PIN1 on its large central cytoplasmic loop[32]. PINOID protein has been shown in yeast two-hybrid assays to bind in a calcium-dependent fashion to two different calcium-binding proteins, TOUCH3 (TCH3 is rapidly induced by mechanical stimuli[33]), and PINOID BINDING PROTEIN 1 (PBP1[34]). Both TCH3 and PBP1 are members of gene families, TCH3 being related to the many calmodulins and calmodulin-like proteins encoded in the *Arabidopsis* genome, and PBP1 to multiple calcium-dependent protein kinases. One hypothesis is therefore that calcium transients in the cytoplasm act to facilitate binding of proteins to PIN1, at least one of which can phosphorylate PIN1 so as to motivate trafficking.

The experiments reported also reveal a new collection of calcium signals in the plant stem cell niche, including mechanically induced slow intercellular calcium waves, spontaneous spikes, and periodic oscillations (as also seen in animal tissues[25]) in the entire inflorescence meristem. The observed mechanical $Ca^{2+}$ response could be mediated by a variety of different components. The initial signal could depend upon plasma membrane localized $Ca^{2+}$ permeable channels, such as those of the CNGC family[16], the GLR family[35–38], the MCA family[39], the OSCA family[40,41], or by the DEK1 protein[42] or TPC1[36,43]. The intercellular calcium waves may in addition rely upon plasmodesmata for $Ca^{2+}$ cell-to-cell diffusion[37], and $Ca^{2+}$ pumps, such as the $Ca^{2+}$-ATPases for cytoplasmic $Ca^{2+}$ level return to the resting state[44].

Spontaneous $Ca^{2+}$ increases have been described previously in plant guard cells by plasma membrane potential hyperpolarization and abscisic acid interaction[26,45]. A transient calcium increase causes short-term stomatal closure, while long-term steady-state stomatal closure is encoded through defined $[Ca^{2+}]_{cyt}$ oscillations[46,47]. Calcium oscillation is also a signal responding to symbiotic microorganism colonization in plant cell nuclei[48]. Furthermore, $Ca^{2+}$ oscillations are critical during the entire fertilization process after gametophyte physical interactions in plants and animals[49–51]. One open question is the origin of the spontaneous spikes, and of the spontaneous oscillations in $Ca^{2+}$ level seen in the SAMs. It could well be that these represent changing mechanical interactions of the meristem cells, with the spikes representing local mechanical interactions due to local cell expansion (which itself is expected from changing auxin levels[6]). The regular oscillations, which occur every 10–30 min, with a fairly regular frequency, remain a mystery in regard to their origin. This study indicates their potential importance for PIN1 protein dynamic polarized movement during growth. It is tempting to imagine that they are related to some regular mechanical oscillation that characterizes the growth of meristems or their cells, such as those that create circumnutation in roots and stems, which is affected by mechanical inputs[52,53]. Micronutation of *Arabidopsis* hypocotyls has a period of 20–60 min[54], similar to the calcium oscillation frequency.

## Methods

**Chemical treatments**. Stock solutions of 100 mM LaCl$_3$ (Sigma-Aldrich) and 100 mM BAPTA (pH 5.8 KOH; Sigma-Aldrich) were prepared in water. In PIN1-GFP growth experiments, for LaCl$_3$ treatment, 5 mM solution was applied for 2 min each hour for 12 h to dissected SAMs embedded at the basal end in a 1% agarose plate. During treatment, SAMs were incubated on GM (Growth Medium, containing 1% sucrose, 1X Murashige and Skoog salts (Sigma M5524), MES 2-(MN-morpholino)- ethane sulfonic acid (Sigma M2933) brought top pH 7 with 1M KOH, plus 0.8 % Bacto Agar (Difco), 1% MS vitamins (Sigma M3900) agar plus 500 nM 6-benzylaminopurine (BAP). The SAMs were thoroughly washed in water to remove LaCl$_3$ when starting sample recovery. The SAMs were then incubated on GM + BAP plates for 12 h. Microscopy was performed immediately before and 12 h after LaCl$_3$ treatment, and 12 h after recovery. Control samples experienced the same treatment steps except that the solution used was water rather than 5 mM LaCl$_3$. In PIN1-GFP growth experiments, for BAPTA treatment, the procedure was similar to LaCl$_3$ treatment, except that 2 mM BAPTA was used, SAMs were

incubated on GM agar plus 500 nM BAP without 3 mM CaCl$_2$ and washed in 3 mM CaCl$_2$ for recovery.

In PIN1-GFP ablation experiments, 5 mM LaCl$_3$ solution was applied to the SAM in the same way as in growth tests, but for 15 min before cell ablation. Water served as the control treatment. After ablation, the SAMs (without water rinse) were directly transferred to GM plates to incubate for 3 h. During recovery, SAMs from which the LaCl$_3$ was washed with water were incubated for an additional 3 h on GM plates. Imaging was performed at 0 and 3 h after ablation, and after 3 h of recovery. For BAPTA treatment, 2 mM BAPTA solution was applied to the dissected SAM for 10 min before cell ablation. After ablation, the SAMs were briefly rinsed with water and then transferred to a 1% agarose plate to incubate for 3 h. During recovery, the SAMs were washed with a solution of 3 mM CaCl$_2$ and incubated for an additional 3 h on GM plates. Imaging was performed at 0 and 3 h after ablation, and after 3 h of recovery.

In MBD-GFP experiments, inhibitor treatments were the same as in the PIN1-GFP experiments, except that the incubation time extended to 6 h, after which imaging was done, and recovery experiments were not performed.

For R-GECO1 and R-GECO1; 29–1-GFP double reporter experiments, LaCl$_3$ or BAPTA treatment was the same as for other reporters, except the test for blockage of spontaneous $Ca^{2+}$ reactions (spikes and oscillations) used 0.5 mM LaCl$_3$ and 0.2 mM BAPTA immersion during one-hour imaging. During R-GECO1 recovery experiments, LaCl$_3$− or BAPTA-pretreated SAMs were cell ablated with a pulled glass pipette and incubated for 3 h on a GM plate (for SAMs with LaCl$_3$ treatment) or on a 1% agarose plate (for SAMs with BAPTA treatment). The sample was then placed under the confocal microscope objective and laser scanning was started immediately after adding water.

**Mechanical perturbation procedures**. For PIN1-GFP and MBD-GFP reporter lines, cellular ablations were performed manually with a pulled glass pipette under a Zeiss SteREO Discovery V8 stereomicroscope. Cellular ablations on the SAMs expressing *pPIN1::PIN1-GFP* and *pML1::mcherry-MAP4* were carried out on a Leica SP5 upright confocal microscope using a multiphoton laser from Spectra-physics which was controlled by Leica software[55]. Ablations were performed using 800 nm wavelength with an output power of ~3 W. Each pulse was shot for 1–15 ms, which varied from sample to sample. Ablations were usually accomplished within 1–3 bursts of the laser.

For observations of R-GECO1 reporter lines, mechanical perturbation (usually accompanied by cell injury/wounding) was performed from the side of the SAM using a pulled glass pipette attached to a micromanipulator. Pipettes were fabricated by pulling a 1-mm diameter glass capillary on a micropipette puller (Sutter Instruments, P-97). The tip diameters of the micropipettes ranged between 3 and 5 μm, about the size of one meristematic cell. The micropipettes were mounted on a three-axis micromanipulator (either Narishige Hydraulic Micromanipulator MO-202 or WPI M3301L manual micromanipulator) allowing precise micropipette movement under the microscope near the SAM (Supplementary Fig. 6a). Mechanical perturbation (wounding) was performed while observing the SAMs with a Zeiss LSM 780 laser scanning confocal microscope. For GCaMP6f response to mechanical perturbation (wounding), cellular ablations were performed in the FRAP mode using a multiphoton laser from MaiTai (Spectra Physics) that was controlled using the Leica software. A single pulse of 800 nm was used for 100 ms to 2 s (time for ablation varied for each plant) using bleach point mode in two-dimension (xyt mode).

For the R-GECO1 SAM compression experiments, the SAM upper surface was pressed and released by a glass coverslip. The device holding the coverslip consisted of a mechanical part made with stainless steel that is open in its center. The coverslip was attached (with vacuum grease) to the part in order to close the open area. The part is mounted on a three-axis micro-manipulator (Narishige Hydraulic Micromanipulator MO-202) (Supplementary Fig. 6b).

**Confocal imaging of fluorescent reporters in living plants**. PIN1-GFP and MBD-GFP fluorescent reporters were imaged using a Zeiss LSM 510 Meta confocal microscope or a Zeiss LSM 780 laser scanning confocal microscope. R-GECO1 and double fluorescent reporters R-GECO1; 29–1-GFP were imaged by using a Zeiss LSM 880 or Zeiss LSM 780 laser scanning confocal microscope. PIN1-GFP and MBD-GFP laser and filter settings were used as described previously[1,7]. Briefly, we used a 488 nm laser for excitation and 505–550 nm for emission collection. To image R-GECO1 and 29-1-GFP simultaneously in the SAM, the multi-tracking mode in the ZEISS LSM 780 was used. R-GECO1 was excited with 561 nm and its emission was detected between 570 and 640 nm. 29-1-GFP was excited with 488 nm and its emission was collected between 500 and 550 nm. Images were recorded using a W Plan-APOCHROMAT ×40 water-dipping objective (NA 1.0) with a time interval of 1 s for $Ca^{2+}$ wave observation (2D), 1 or 2 s for $Ca^{2+}$ spike observation (2D), 5 s for $Ca^{2+}$ spike observation (3D), and 5 or 10 s for $Ca^{2+}$ oscillation observation (2D). The movie was recorded in 512 × 512 pixels format. The intervals between optical slices in z-stacks ranged from 1 to 2 μm. Confocal z-stack images were processed using ImageJ (imagej.nih.gov/ij/).

The inflorescence meristems of plants carrying GCaMP6f were dissected and mounted as above. The meristems were incubated for ~15 min to minimize calcium response from the dissections. Spikes were captured using a resonance scan mode on Leica TCS-SP5 upright microscope at a scan speed of 8000 Hz. The movie

was recorded in $512 \times 512$ pixels format. For recording GCaMP6f response to mechanical perturbation (wounding), the imaging was performed on a Leica TCS-SP5 upright confocal laser-scanning microscope using a ×25 water-dipping objective (NA-0.95). Five frames were recorded prior to laser ablation and 100 frames were captured immediately after laser-induced ablation at a scan speed of 400 Hz and a scan rate of one optical slice per 0.7 s.

The meristems carrying pPIN1::PIN1-GFP and pML1::mcherry-MAP4 reporters were imaged on a Leica SP5 TCS microscope. PIN1-GFP was excited using 488 nm argon laser and mcherry was excited using 561 nm white light laser.

The SAM oscillation signal observation in intact plants was performed using an imaging box (Cat. #64313, Electron Microscopy Sciences). Plants were initially grown on soil in small pots until they bolted, then together with the pots, they were transferred into imaging boxes. The flower-dissected SAM was isolated from leaves and roots by placing Parafilm around the stem and sealing with a thin layer of 1% agarose to prevent water leakage. Water immersed SAMs were located under a ×40 objective for signal observation.

**Code availability**. The corresponding open-source Python code, previously introduced in Refs. [56,57] on natural-cubic-smoothing-splines can be found at https://github.com/eldad-a/natural-cubic-smoothing-splines. All other code, including the programming code for quantitative analysis of the meristem calcium oscillations and waves upon mechanical perturbation, are available from the corresponding authors upon request.

**Reporting summary**. Further information on experimental design is available in the Nature Research Reporting Summary linked to this article.

## Data availability

All data are available from the corresponding author upon request, including all of the microscope time lapse files. The source data underlying Supplementary Figs. 1a, b, 4c, 5a–f, 7e–j, 8a, b, 10, 11i–l, 12k–l and Supplementary Tables 1 and 2 are provided as a Source Data file.

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

## Acknowledgements

We are grateful to Dr. Arun Sampathkumar for support in establishing the calcium confocal imaging process, Dr. Hanako Yashiro for the support in intact plant imaging and Dr. Ivo Grosse for the useful discussions and input on statistical analysis. We thank Dr. Francesca Peri for kindly providing the DNA for GCaMP6f and Dr. Rainer Waadt for R-GECO1 constructs and advice. We also thank Dr. Changfu Yao and Meyerowitz and Heisler lab members for suggestions and discussions. The authors' work was funded by the Howard Hughes Medical Institute and by NASA grant NNX17AD53G to E.M.M., by the European Research Council (ERC) grant 261081 to M.G.H., in part by NIH grant GM060396 to J.I.S., Marie Skłodowska-Curie Action fellowship to P.D-.S., and a California Institute of Technology Biology and Biological Engineering Fellowship and a Zuckerman STEM Leadership Program postdoctoral scholarship to E.A.

## Author contributions

T.L., A.Y., N.B., J.I.S, M.G.H., and E.M.M. conceived the experiments. T.L., N.B., A.Y., P.D-.S. performed experiments. A.A. analyzed $Ca^{2+}$ wave imaging data. E.A. analyzed $Ca^{2+}$ oscillation patterns. P.D-.S. invented, built, and provided the indentation and compression devices. P.T.T. contributed reagents. T.L., A.Y., N.B., M.H., and E.M.M. wrote the manuscript. All authors read and approved the manuscript.

## Additional information

**Competing interests:** The authors declare no competing interests.

