## [Peer Review File · Nature Communications]

Reviewers' comments:

Reviewer #1 (Remarks to the Author):

The importance of PIN1 polarization for phyllotactic patterning is well established, but how PIN1 polarization is controlled at the molecular level remains an important question. This exciting study reveals a role for cytosolic Ca²⁺ in patterning PIN1 at the shoot apical meristem. Using fluorescence microscopy approaches, the authors observed cytosolic Ca²⁺ concentration changes in the SAM not only in response to external mechanical perturbation but also during normal growth. Pharmacological inhibition of Ca²⁺ increases abolished the formation of new flower primordia and changes in PIN1 polarity, without affecting mechanically induced microtubule reorientation. Overall, the experiments in this study are carefully performed and interpreted, especially with regard to the investigation of Ca²⁺-dependent repolarization of PIN1 after mechanical stimulation. However, I think the manuscript would be improved by more thorough quantitative analysis of the data and a few additional control experiments.

1. Ca²⁺ oscillations in growing SAMs:

The observation of Ca²⁺ oscillations during normal development of SAMs is fascinating and deserves more in-depth analysis:

(i) Do these oscillations occur in intact plants (non-excised SAMs)? GCaMP6f measurements should be possible in intact inflorescence stems and would confirm that these oscillations are not an artefact of sample preparation.

(ii) In Supplemental Video 1, the oscillations appear to propagate across the SAM and then the primordia/young leaves. Is the velocity similar to the velocity in mechanically triggered Ca²⁺ waves? Are oscillations typically initiated in the same region of the SAM and do these sites predict new organ initiation?

(iii) La³⁺ blocks Ca²⁺ oscillations but also appears to completely inhibit growth (not just the development of new primordia). Some attempt to quantify growth effects should be made and it should be shown that BAPTA has similar effects on primordia formation/growth and endogenous Ca²⁺ oscillations.

(iv) Effects of La³⁺ on PIN1 polarization pattern should be quantified (e.g. how many PIN1 maxima disappear/newly appear/change intensity in control SAMs versus La³⁺ treated SAMs?). Does BAPTA also cause an increase in PIN1-GFP signal?

2. Mechanically induced PIN1 repolarization:

Endogenous Ca²⁺ transients oscillate but each individual transient is fairly brief. Is a single mechanical (non-wounding) stimulus sufficient to trigger PIN1 repolarization? This would address the question of how sustained Ca²⁺ increases (along with other unknown signals) must be to trigger PIN1 polarization. Are Ca²⁺ increases sufficient to repolarize PIN1?

3. Microtubule reorientation:

It is stated that microtubule reorientation in response to mechanical perturbation is not affected by La³⁺/BAPTA treatment (unlike PIN1). This is an exciting result that should be better supported by a quantitative analysis (e.g. PIN1/MT correlation analysis as in Heisler et al 2010)

4. PIN1 polarization recovery in mechanically stimulated SAMs after La³⁺ washout.

(i) It was not clear to me whether the Ca²⁺ increase after La³⁺ washout in wounded SAMs was due to the previous wounding stimulus or due to the washing treatment, especially given that the kinetics of the Ca²⁺ transients appear different (monophasic versus biphasic). A control washout experiment using non-wounded, La³⁺ treated SAMs would address this.

(ii) Does the Ca²⁺ increase upon washout also occur in SAMs that were mechanically stimulated but not wounded?

Minor points:

- Figure 2r-u: a brief reminder in the legend that red/blue corresponds to high/low DII/mDII ratios, reflecting low/high auxin levels would be helpful to readers not familiar with this sensor.
- p. 5, line 111: Were samples treated with 10fold dilutions of La³⁺/BAPTA for these experiments (why?), or is this a spelling error?
- The text would be easier to follow if the authors clearly differentiated between mechanically wounded versus mechanically stimulated SAMs
- Nakayama et al (2012) Current Biology 22: 1468 should be cited (plasma membrane tension affects PIN1 localization).

Reviewer #2 (Remarks to the Author):

The manuscript by Li, Yan, Bhatia and co-workers provides a very interesting description of fascinating dynamic Ca²⁺ signals in the shoot apical meristem occurring spontaneously and in response to mechanical stimuli. The Ca²⁺ wave propagation after mechanical stimulation was proposed to underlie the initiation of PIN repolarization, independently of microtubule reorganization.

Whereas the manuscript contains many interesting observations, they are often not really connected to each other, and sometimes overinterpreted.

The main cause for this is that most of the causal connections that were proposed are inferred from the use of the Ca²⁺ chelator BAPTA and the channel blocker LaCl₃. Both treatments having many non-specific effects, and possibly also causing Ca²⁺ unrelated effects at the concentrations at which they were used. This is recognized by the authors, as they often use wash-out experiments to demonstrate that the treated samples were still alive. Yet, this does not demonstrate specificity. This is also suggested by the observations that they could block the spontaneous Ca²⁺ dynamics by 10x lower concentrations BAPTA and LaCl₃ than those used for disturbing mechanically induced Ca²⁺ signals.

The induction of a wave of Ca²⁺ signals after mechanical stimulation demonstrates cell-to-cell signal propagation. Thus far, Ca²⁺ wave propagation (salt, herbivory, aphid feeding) has been described to depend on BAK1, GLR3.3/GLR3.6, TPC1 function, ROS (Choi et al. 2014 PNAS, Evans et al., 2016 Plant Phys, Kiep et al. New Phytol., Vincent et al. 2017 Plant Cell). Moreover, mechanical sensing in plants was connected to the receptor-like kinase FERONIA (Shih et al. 2014 Curr Biol), and a touch-sensitive Ca²⁺ channels has been characterized (MCA1). This demonstrates that a quite some possible, testable molecular mechanisms are out there to benchmark their observations.

Minor comments:

- the legends are a bit superficial, and thus often do not provide sufficient details about how an experiment was done. Concentrations, timings, ...
- PIN polarity changes in Fig1 in the control are not very clear, and might be clearer by inclusion of arrow heads that indicate polarity.
- What happens to auxin (re)distribution after BAPTA treatment?
- In Fig2, it is clear that LaCl₃ treatment has a general effect on R2D2 ratios, also in primordia. Does this reflect a general auxin redistribution, or an effect on TIR/AFB based auxin signaling? The latter is important as TIR1/AFB was recently connected to Ca²⁺ signaling, which in turn controlled auxin-regulated expression of IAA19 (Dindas et al., 2018; Nat Commun)
- The 5mM LaCl₃ and 2mM BAPTA are very high. What is the minimal concentration to disrupt mechanically induced Ca²⁺ signals.
- If LaCl₃ causes an increased PIN levels in Fig1, how can this be reconciled with BAPTA lowering PIN levels (Extended Data Fig1)?
- What are the Ca²⁺ dynamics after laser ablation of cell in the SAM? Now Ca²⁺ waves are

described by prodding the meristem, and are connected to PIN polarization after cell ablation.
- The last experiment shows that a LaCl₃ or BAPTA treatment after mechanical stimulation cannot prevent PIN relocation. However, it is unclear how long it takes for the inhibitors to penetrate the tissues sufficiently to impair Ca²⁺ signaling. This can be important information for correctly interpreting this experiment.

Reviewer #3 (Remarks to the Author):

In this manuscript Li et al explored the role of Ca²⁺ signaling in PIN1 polarization and subsequently its impact on auxin transport in the shoot apical meristem. Role of Ca²⁺ has been explored extensively in cell polarity from lower forms to higher forms of organisms. A large body of work devoted by the authors in the last decade to understand the role of PIN1 in auxin transport in shoot apex and how it gets polarized laid down a framework and many labs around the globe successfully used the resources and knowledge generated by them to characterize various mutants.

In this manuscript, the authors conducted a series of experiments to show that Ca²⁺ flux is generated within the shoot apical meristem cells in a periodic fashion evident from the live imaging experiments. This is generated perhaps via the release of Ca²⁺ ions in these cell types though it has no apparent correlation with any preferential PIN1 localization. By blocking the Ca²⁺ release the authors show that PIN1 polarization too get locked in the cells where it was initially expressed. This also has an impact on organogenesis. This observation sounds well and reasonable to expect if Ca²⁺ signaling plays such an important role. By these experiments authors conclude that the free Ca²⁺ ions availability is essential for normal functioning of shoot apical meristem because the PIN1 is also needed for normal phyllotaxy and auxin transport.

The authors also tested the possibility whether Ca²⁺ has any impact on microtubule orientation, which the authors standardized in the past in their laboratory. Based on the mechanical perturbation and needle injury experiments, the authors conclude that in the presence of LaCl₃, MT orientation does not get perturbed, however, the PIN1 polarization get affected. Suggesting that LaCl₃ blocks specifically PIN1 polarization and not act as a general inhibitor of ion transport. Hence PIN polarization and Ca²⁺ release is somehow linked and need to be dissected out to understand its precise role in cell polarity.

The authors could show with the available resources and tools that there is burst of Ca²⁺ in meristem, and does indeed influence the PIN1 polarization. The findings presented in this study are novel, and would certainly help the wider community to appreciate the role of Ca²⁺ in meristematic cells.

However, the authors need to address following things.

Major comments

Despite the fact that LaCl₃ blocks calcium release, how specific it is and how to rule out the possibility that other channels related to Ca²⁺ signaling are not getting affected by this treatment. This aspect needs to be addressed in this article.

There are reports where LaCl₃ was also shown to block K⁺. The authors only focused on the PIN1 and strongly argue without testing other PIN transporter proteins.

It is evident from the LaCl₃ treatment that treated plants do not show a PIN-like shoot phenotype.

This study is still inconclusive from this reviewer's perspective. It explored the role of Ca²⁺ signaling in cell polarity but there is no evidence yet that suggest that Ca²⁺ get released in cell type specific manner and dictate the PIN1 polarity ultimately. Experiment involving genetics can decouple the role of Ca²⁺ signaling in PIN1 polarization.

Minor comments

Why the authors choose to do the cortical microtubule orientation experiment test using 5mM LaCl₃, whereas their other observation recorded at 3mM, if there are any specific reasons for this then it need to be addressed in the text.

Reviewer #1 (Remarks to the Author):

The importance of PIN1 polarization for phyllotactic patterning is well established, but how PIN1 polarization is controlled at the molecular level remains an important question. This exciting study reveals a role for cytosolic Ca²⁺ in patterning PIN1 at the shoot apical meristem. Using fluorescence microscopy approaches, the authors observed cytosolic Ca²⁺ concentration changes in the SAM not only in response to external mechanical perturbation but also during normal growth. Pharmacological inhibition of Ca²⁺ increases abolished the formation of new flower primordia and changes in PIN1 polarity, without affecting mechanically induced microtubule reorientation. Overall, the experiments in this study are carefully performed and interpreted, especially with regard to the investigation of Ca²⁺-dependent repolarization of PIN1 after mechanical stimulation. However, I think the manuscript would be improved by more thorough quantitative analysis of the data and a few additional control experiments.

1. Ca²⁺ oscillations in growing SAMs:

The observation of Ca²⁺ oscillations during normal development of SAMs is fascinating and deserves more in-depth analysis:

(i) Do these oscillations occur in intact plants (non-excised SAMs)? GCaMP6f measurements should be possible in intact inflorescence stems and would confirm that these oscillations are not an artefact of sample preparation.

Response: We thank the reviewer for the comments. We agree that it is critical to demonstrate that oscillations do not only occur in excised SAMs, but also in intact plants as well. We have now imaged intact plants and observed that the oscillation signal inter-peak time and Full-Width Half Maximum time is similar to that of excised SAMs (we added these data into our manuscript in Supplementary Figure 5 c-f and supplementary Movie 2). We also found the oscillation signal peak could become weaker when keeping imaging longer than about two hours for intact plant observations, indicating a possibility that the Ca²⁺ signal is, over long time periods, partially repressed by tissues outside the SAM under the current intact plant imaging condition. Based on this reasoning, we therefore

suspect that imaging excised SAMs provides a better indication of signals generated in the SAM.

(ii) In Supplemental Video 1, the oscillations appear to propagate across the SAM and then the primordia/young leaves. Is the velocity similar to the velocity in mechanically triggered Ca²⁺ waves? Are oscillations typically initiated in the same region of the SAM and do these sites predict new organ initiation?

Response: Regarding the velocity of oscillations compared to mechanically triggered Ca²⁺ waves, the reviewer raises a very interesting point that we did not address in the initial manuscript. This point is well worth investigation. To directly compare these two signals, we performed a new set of experiments using a single SAM to image both the oscillation signal and mechanically triggered waves sequentially with the same imaging settings (oscillation was recorded first, and then the Ca²⁺ wave during mechanical perturbation followed). After side by side comparison, we found that when spreading from the same 2D area, the time for the oscillation signal to progress across the meristem was 8.8 ± 4.4 (mean \pm SD, $n = 9$ individual SAMs) fold the time for mechanically triggered Ca²⁺ waves. In other words, these two velocities are very different, and the oscillation signal propagation velocity is much slower than the Ca²⁺ wave by mechanical perturbation. We added this statement in the main text on page 10.

In addition, we compared the Ca²⁺ sensor R-GECO1 fluorescence intensity changes in these two different situations and found the amplitude and duration of the two signal peaks are also different. The maximum intensity fold change ($(I - I_0)/I_0$) of the mechanically induced Ca²⁺ wave is about 2.3 ± 1.1 (mean \pm SD, $n = 7$); the one for the oscillation signal is 0.5 ± 0.2 (mean \pm SD, $n = 12$). The duration of the major peak of the mechanically induced wave is less than 2 min, but the oscillation signal duration takes longer than 10 min on average. These data were added to the main text on page 11 and as a new Supplementary Figure 10.

Regarding the second question about the region of oscillation initiation, we re-examined our oscillation movies, the data do not support the hypothesis that the oscillations are initiated in the same region, such as a new organ initiation region. So the origin of oscillation is still a mystery. We included a representative movie as supplemental movie 2.

(iii) La³⁺ blocks Ca²⁺ oscillations but also appears to completely inhibit growth (not just the development of new primordia). Some attempt to quantify growth effects should be made and it should be shown that BAPTA has similar effects on primordia formation/growth and endogenous Ca²⁺ oscillations.

Response: To address this issue, we decided to monitor growth by tracking epidermal cell divisions. We quantified the proportion of cells that divided over a 12h period based on total SAM cell number in the epidermis from ten mock samples and ten LaCl₃ -treated samples and found that for each LaCl₃ treated sample, cell division still occurred but at a lower frequency compared to mock (plot is shown as Supplementary Fig. 1a).

We have also now included a similar growth analysis for a PIN1-GFP reporter line under BAPTA treatment as well. We found that the BAPTA had a similar effect on cell division rates although they were a little lower compared to LaCl₃ (plot is shown as Supplementary Fig. 1b). We also found that primordium formation and growth was slowed, although there is no PIN1-GFP fluorescent signal increase as there is during LaCl₃ treatment ($n = 10$ for BAPTA-treated SAMs, $n = 6$ for mock treated SAMs. We added images for these results in a new Supplementary Figure 2).

Regarding the BAPTA effect on endogenous Ca²⁺ oscillations, we applied 0.2mM BAPTA to SAMs and observed R-GECO1 signals for 1h. We found that like LaCl₃, BAPTA also blocked oscillation signals (please check the supplementary Movie 6).

(iv) Effects of La³⁺ on PIN1 polarization pattern should be quantified (e.g. how many PIN1 maxima disappear/newly appear/change intensity in control SAMs versus La³⁺ treated SAMs?). Does BAPTA also cause an increase in PIN1-GFP signal?

Response: Mock treated plants formed at least one new convergence of PIN1-GFP protein surrounding newly emerged floral primordia ($n = 16$), but no new PIN1-GFP convergences appeared in LaCl₃ -treated SAMs ($n = 10$) or BAPTA-treated SAMs ($n = 10$). For better visualization, we highlighted with colored dots the cells

that had PIN1 polarity changes in mock treated samples and the cells that had no PIN1 polarity changes in LaCl_3 or BAPTA treated samples, but proceeded to change during later recovery (Figure 1 and the Supplementary Figure 2). Unlike La^{3+} , we did not see an increase in PIN1-GFP signal in response to BAPTA.

2. Mechanically induced PIN1 repolarization:

Endogenous Ca^{2+} transients oscillate but each individual transient is fairly brief. Is a single mechanical (non-wounding) stimulus sufficient to trigger PIN1 repolarization? This would address the question of how sustained Ca^{2+} increases (along with other unknown signals) must be to trigger PIN1 polarization.

Response: We think a single transient mechanical (non-wounding) stimulus is not sufficient to trigger PIN1 repolarization since the repolarization happens over several hours and a sustained mechanical stimulus (such as in the cell ablation case) is necessary to trigger the repolarization. Currently it is beyond our technical ability to apply sustained non-wounding localized mechanical pressure to the SAM, but we will work hard to address this question in future studies.

Are Ca^{2+} increases sufficient to repolarize PIN1?

Response: Our conclusion in general is that mechanical stress controls PIN1 and that Ca^{2+} increases are necessary but not sufficient for repolarization. This makes sense as the reaction time scales of Ca^{2+} oscillations and PIN1 repolarization are different, and Ca^{2+} , as a second messenger, responds to all kinds of environmental simulation. We clarified our conclusion in the main text abstract (page 2).

3. Microtubule reorientation:

It is stated that microtubule reorientation in response to mechanical perturbation is not affected by La^{3+} /BAPTA treatment (unlike PIN1). This is an exciting result that should be better supported by a quantitative analysis (e.g. PIN1/MT correlation analysis as in Heisler et al 2010).

Response: Thanks to the reviewer for the suggestion. We agree that the microtubule reorientation and PIN1 intensity maxima should be analysed

quantitatively to better support our statement. We have generated a new Supplementary Figure 4 to present this analysis and have revised our text.

Since in La^{3+} pre-treated plants, most of the PIN1-GFP signal was localized on the lateral membranes four hours after mechanical perturbation, unlike the mock treated plants, we could not perform a similar analysis to that reported in (Heisler et al, 2010). Therefore, we quantified cortical microtubule orientations and estimated PIN1-GFP signal localisation separately. This is shown in Supplementary Figure 4.

4. PIN1 polarization recovery in mechanically stimulated SAMs after La^{3+} washout.

(i) It was not clear to me whether the Ca^{2+} increase after La^{3+} washout in wounded SAMs was due to the previous wounding stimulus or due to the washing treatment, especially given that the kinetics of the Ca^{2+} transients appear different (monophasic versus biphasic). A control washout experiment using non-wounded, La^{3+} treated SAMs would address this.

Response: This is an interesting question. We noticed that the wave can be initiated away from the wound (see Supplementary Movie 11 and 12, in which signal increase initiation during LaCl_3 and BAPTA washout is not specifically initiated around the wounding area), thus we suspect that the increases are not due to the previous wounding stimulus but rather the washing treatment. But considering that our hypothesis is that any type of Ca^{2+} increase will allow PIN1 to respond, we think this experiment does not help us to further test our hypothesis.

(ii) Does the Ca^{2+} increase upon washout also occur in SAMs that were mechanically stimulated but not wounded?

Response: This is another interesting question, but as in the response to the comment above, we do not think Ca^{2+} increase is a delayed response to the wounding stimulus. The Ca^{2+} increase after washing is not directly related to the previous mechanical stimulus, wounded or non-wounded.

Minor points:

- Figure 2r-u: a brief reminder in the legend that red/blue corresponds to high/low DII/mDII ratios, reflecting low/high auxin levels would be helpful to readers not familiar with this sensor.

Response: A comment raised by another reviewer questions the R2D2 sensor validity for reading the auxin redistribution when adding LaCl_3 based on a recent report that shows that LaCl_3 treatment inhibits TIR1/AFB signaling in roots (Dindas et al., 2018). We performed an additional test to re-examine if it is the same in the SAMs, and it appears that it is.

In the original manuscript, we demonstrated that Ca^{2+} signals are only necessary for the initiation of the PIN1 protein repolarization after mechanical perturbation, and are not required for later PIN1 protein trafficking (Fig. 4). We performed the similar time shift treatment (ablated cells first, waited five minutes, then treated with 5mM LaCl_3 for 15 min, and retreated the samples in 5mM LaCl_3 for 2min every 2h during incubation on GM plates till 6h) on R2D2 sensor.

We hypothesize that if the LaCl_3 treatment inhibits the TIR1/AFB signaling pathway in SAMs, as reported by Dindas et al., 2018, and in this experiment, LaCl_3 is supposed to block the majority of Ca^{2+} signals in the 6h duration except the first Ca^{2+} wave in response to mechanical perturbation, R2D2 should not respond to the auxin redistribution after PIN1 repolarization in this time shift experiment (Fig. 4). Otherwise, if the auxin-interaction domain in DII-VENUS is not functionally disturbed by LaCl_3 , R2D2 could still read the auxin redistribution after PIN1 repolarization under the time shifted LaCl_3 treatment, and its fluorescence ratios would show similar changes as the samples with mock treatment, as shown by PIN1-GFP protein responses. Our testing results are consistent with the first hypothesis that the TIR1/AFB signalling pathway, or at least R2D2's AUX/IAA based auxin signalling element, is indeed partially inhibited by LaCl_3 , as we observed the R2D2 DII/mDII ratio changes were inhibited at certain level in this time shift experiment. This response is also different when comparing to the response of mock samples and is inconsistent with PIN1 repolarization behaviour under similar treatment. Overall our data do agree with the conclusions of Dindas et al. ($n = 8$, the figure is attached below).

a-d, Rainbow color-coded ratiometric images derived by dividing signal in the DII-n3xVenus channel by that in the mDII-ntdTomato channel of R2D2 at 0h and 6 h after pipette-induced cell ablation (marked as dashed circle) without (**a, b**) or with (**c, d**) 5mM LaCl₃ posttreatment.

In short, we agree that R2D2 is not a practicable sensor in our current study to demonstrate the Ca²⁺ signal effect on auxin distribution through the requirement on PIN1 repolarization due to the complexity of Ca²⁺ signal functions on both auxin signalling and auxin transport. We therefore decided to remove panel n-u from Fig. 2 to avoid any misleading conclusions; the experiment was never necessary to support our conclusions.

For this one we really have to thank the reviewer, as we had not yet seen the Dindas et al paper.

- p. 5, line 111: Were samples treated with 10 fold dilutions of La³⁺/BAPTA for these experiments (why?), or is this a spelling error?

Response: It is a 10-fold dilution, not a spelling error. As the strength of the Ca²⁺ response to mechanical perturbations appeared quite strong compared to the responses we had observed earlier, we wanted to determine the optimal concentration of La³⁺/BAPTA for inhibition of this response. This can be seen by the Ca²⁺ response level comparison in Ca²⁺ intensity change and propagation time during Ca²⁺ wave and oscillation (See the above response to the major comment 1(ii)) and data are available to the main text on page 11 and as a new Supplementary Figure 10.

In addition, for blocking the Ca²⁺ mechanical response, we did a short time (10-15 min) pre-treatment and for blocking Ca²⁺ oscillations, we immersed samples for another 1h during time lapse imaging. To avoid any unnecessary side effects from long-term high-concentration drug treatments, we adopted 0.5mM instead of 5mM LaCl₃.

- The text would be easier to follow if the authors clearly differentiated between mechanically wounded versus mechanically stimulated SAMs

Response: Thanks for the comments and we have revised them in the manuscript.

- Nakayama et al (2012) Current Biology 22: 1468 should be cited (plasma membrane tension affects PIN1 localization).

Response: The reference is cited in the text (page 13). We thank the Reviewer 1 for pointing out this. We would point out that what Nakayama et al. mean by localization (partitioning between plasma membrane and cytoplasmic vesicles) and what we mean (asymmetric distribution of PIN1 within the plasma membrane) are different, and we make this explicit at the start of the paper now.

Reviewer #2 (Remarks to the Author):

The manuscript by Li, Yan, Bhatia and co-workers provides a very interesting description of fascinating dynamic Ca²⁺ signals in the shoot apical meristem

occurring spontaneously and in response to mechanical stimuli. The Ca^{2+} wave propagation after mechanical stimulation was proposed to underlie the initiation of PIN repolarization, independently of microtubule reorganization.

1. Whereas the manuscript contains many interesting observations, they are often not really connected to each other, and sometimes overinterpreted. The main cause for this is that most of the causal connections that were proposed are inferred from the use of the Ca^{2+} chelator BAPTA and the channel blocker LaCl_3 . Both treatments having many non-specific effects, and possibly also causing Ca^{2+} unrelated effects at the concentrations at which they were used. This is recognized by the authors, as they often use wash-out experiments to demonstrate that the treated samples were still alive. Yet, this does not demonstrate specificity.

Response: We agree that pharmacological tests always come up with non-specific issues that are hard to avoid, this is the main reason we chose two different Ca^{2+} related chemicals to test our hypothesis. These two treatments work through entirely different mechanisms. Each of LaCl_3 and BAPTA may have non-specific effects, but they are not expected both to have the same non-specific effects, so that the common response to both should be the calcium response. And the experiment of wounding prior to Lanthanum or BAPTA where PIN1 re-oriented demonstrates that it is not due to an artefact that occurs beyond Ca^{2+} signals (Fig. 4).

This is also suggested by the observations that they could block the spontaneous Ca^{2+} dynamics by 10x lower concentrations BAPTA and LaCl_3 than those used for disturbing mechanically induced Ca^{2+} signals.

Response: Thanks for pointing out this difference. We think that cell ablation is an extreme stimulation for mechanical force change (the turgor pressure adjacent to the cell changes to zero) and is combined with a wounding stress. In contrast, endogenous Ca^{2+} signals are generated from endogenous cell signalling and cell-cell interactions (the exact origin of Ca^{2+} endogenous oscillation and spikes is still a mystery), and so are likely to be weaker.

To directly compare these two signals, we performed a new set of experiments that used a single SAM to image both the oscillation signal and mechanically triggered wave sequentially with the same imaging settings (oscillation was recorded first, and then the Ca²⁺ wave during mechanical perturbation recording followed). After side by side comparison, we found that when spreading over the same 2D area, the time that an oscillation signal spent was 8.8 ± 4.4 (mean \pm SD, $n = 9$ individual SAMs) times the interval of the time for mechanically triggered waves. In other words, these two velocities are different, and oscillation signal propagation velocity is much slower than the Ca²⁺ wave by mechanical perturbation. We added this statement in the main text on page 10.

In addition, we compared the Ca²⁺ sensor R-GECO1 fluorescent intensity changes in these two reactions and found the amplitude and duration of the two signal peaks are also different. The maximum intensity fold change $(I-I_0)/I_0$ of the mechanically induced Ca²⁺ wave is about 2.3 ± 1.1 (mean \pm SD, $n = 7$); the one of oscillation signal is 0.5 ± 0.2 (mean \pm SD, $n = 12$). The duration of the major peak of the wave is in about 2 min, but the oscillation signal duration time is longer than 10 min on average. These data were added to the main text on page 11 and as a new Supplementary Figure 10.

These two Ca²⁺ signal levels are different, Ca²⁺ waves in response to mechanical stimulation is much stronger than spontaneous Ca²⁺ dynamics. This response was also given to reviewer 1, above, who had the same concern.

The induction of a wave of Ca²⁺ signals after mechanical stimulation demonstrates cell-to-cell signal propagation. Thus far, Ca²⁺ wave propagation (salt, herbivory, aphid feeding) has been described to depend on BAK1, GLR3.3/GLR3.6, TPC1 function, ROS (Choi et al. 2014 PNAS, Evans et al., 2016 Plant Phys, Kiep et al. New Phytol., Vincent et al. 2017 Plant Cell). Moreover, mechanical sensing in plants was connected to the receptor-like kinase FERONIA (Shih et al. 2014 Curr Biol), and a touch-sensitive Ca²⁺ channels has been characterized (MCA1). This demonstrates that a quite some possible, testable molecular mechanisms are out there to benchmark their observations.

Response: We agree with the reviewer that there are many potentially interesting links to findings in the current literature. However we feel that such experiments

are suitable for follow up studies. We have now included suggestions for future work in the discussion part of main text and have cited the corresponding references, including the newly published one on GLR3.3/GLR3.6 mediation of cell-to-cell Ca^{2+} signal propagation (Toyota et al. 2018 Science; Nguyen et al. 2018 PNAS) (the main text on page 13).

Minor comments:

- the legends are a bit superficial, and thus often do not provide sufficient details about how an experiment was done. Concentrations, timings, ...

Response: We have now added more of this detail to the manuscript.

- PIN polarity changes in Fig1 in the control are not very clear, and might be clearer by inclusion of arrow heads that indicate polarity.

Response: We agree this could be improved. For better visualization, we have now highlighted, using colored dots, the cells that had PIN1 polarity changes in mock treated samples and the cells that had no PIN1 polarity changes in LaCl_3 or BAPTA treated samples but proceeded to show changes during later the recovery stage (Figure 1 and the Supplementary Figure 1).

-What happens to auxin (re)distribution after BAPTA treatment?

Response: At present, it seems LaCl_3 changes the auxin distribution, as expected from the PIN1 response. However, it also changes TIR1/AFB auxin signalling (See the response of the next comment below). Since the R2D2 sensor, which is the only sensor available for monitoring auxin concentration dynamics at cellular resolution, depends on TIR1/AFB, we decided to eliminate the experiments that relied on R2D2 as an auxin sensor because we can't separate the effects of auxin transport from those on auxin response.

-In Fig2, it is clear that LaCl_3 treatment has a general effect on R2D2 ratios, also in primordia. Does this reflect a general auxin redistribution, or an effect on TIR/AFB based auxin signaling? The latter is important as TIR1/AFB was recently

connected to Ca²⁺ signaling, which in turn controlled auxin-regulated expression of IAA19 (Dindas et al., 2018; Nat Commun)

Response: We appreciate this comment very much that it points out an important issue that was initially not realized by us. This comment prompted us to perform an additional test to re-examine the validity of the R2D2 sensor in assessing the auxin distribution when adding LaCl₃ and/or BAPTA.

In this manuscript, we demonstrated that Ca²⁺ signals are only necessary for the initiation of the PIN1 polarization response to mechanical perturbation and are not required for later PIN1 protein trafficking (Fig. 4). We performed a similar time shift treatment using (ablated cells first, waited five minutes, then treated with 5mM LaCl₃ for 15 min, and retreated the samples in 5mM LaCl₃ for 2min in every 2h during incubation on GM plates till 6h) the R2D2 sensor.

We hypothesized that if the LaCl₃ treatment inhibits the TIR1/AFB signalling pathway in the SAMs, as reported by Dindas et al., 2018, then the 6 hrs of LaCl₃treatment should prohibit R2D2 from changing over this duration (Fig. 4). Alternatively, if this pathway is not functionally disturbed by LaCl₃, the R2D2 marker should indicate auxin distribution changes after PIN1 has repolarized, similar to the changes observed for mock treatment. Our testing results are consistent with the first hypothesis that the TIR1/AFB signalling pathway, or at least R2D2's AUX/IAA based auxin signalling element, is indeed partially inhibited by LaCl₃, as we observed the R2D2 DII/mDII ratio changes were inhibited at certain level in this time shift experiment. This response is also different when comparing to the response of mock samples and is inconsistent with PIN1 repolarization behaviour under similar treatment. Overall our data do agree with the conclusions of Dindas et al. ($n = 8$, the figure is attached below).

a-d, Rainbow color-coded ratiometric images derived by dividing signal in the DII-n3xVenus channel by that in the mDII-ntdTomato channel of R2D2 at 0h and 6 h after pipette-induced cell ablation (marked as dashed circle) without (**a, b**) or with (**c, d**) 5mM LaCl₃ posttreatment.

We thank the reviewer for raising this issue and we agree that R2D2 is not a reliable sensor in our current study to demonstrate changes in the auxin distribution pattern in response to inhibition of Ca²⁺ signalling. We decided to remove panels n-u from Fig. 2 to avoid any misleading conclusions.

- The 5mM LaCl₃ and 2mM BAPTA are very high. What is the minimal concentration to disrupt mechanically induced Ca²⁺ signals.

Response: We thank the reviewer for bringing up this question. In our revised manuscript, we included a dilution series of LaCl_3 and BAPTA concentrations followed by assessment of the effect of treatment on mechanically induced Ca^{2+} signals (Supplementary Figure 8). We found on average LaCl_3 started to block the Ca^{2+} wave propagation at 1 mM concentration, and for BAPTA, 1 mM concentration is also necessary to block the Ca^{2+} waves.

The concentrations of LaCl_3 used on the SAMs are higher than other tissues, such as roots (ranges in hundreds of micromolar), probably because of the impermeable wax layer coating on the SAM surface. The same is true for oryzalin (depolymerizing microtubules) treatment: 170 nM - 1 μM were used for root response threshold (Baskin et al. 1994 Plant Cell Physiol.); 29-58 μM were used for SAM response treatment (Hamant et al. 2008 Science).

In addition, for PIN1-GFP mechanical response experiments, after cell ablation, we did not immerse the SAMs in LaCl_3 or BAPTA throughout the incubation time. Instead, we used residual chemical to sustain the blockage effect with treatments only for a few minutes each hour. Constant treatment over the time course of some of our experiments induces detrimental effects on growth and tissue viability. If the concentrations were minimal for signal blockage, they would be easily diluted out during later incubation time. We therefore used a higher concentration for shorter times.

- If LaCl_3 causes an increased PIN levels in Fig1, how can this be reconciled with BAPTA lowering PIN levels (Extended Data Fig1)?

Response: Regarding the PIN1-GFP fluorescent signal increase upon LaCl_3 treatment, it is an interesting response, but we do not know why it occurs. It is possible that the LaCl_3 had an effect on the PIN1 transcription levels via influencing PIN1 promoter activity (the PIN1 promoter is auxin responsive, and LaCl_3 affects auxin response), or it is possible that LaCl_3 had effects on the overall rates of PIN1 protein trafficking, thereby altering the balance of protein endocytosis and exocytosis.

In Supplementary Figure 2, after 12h of pulsed BAPTA treatment, PIN1-GFP intensity did not change noticeably compared to 0h. Thus, the two treatments

have different effects on PIN1-GFP protein intensity. However, BAPTA is an extracellular calcium chelator while LaCl_3 is a plasma membrane blocker so it is possible the distinct responses in terms of PIN1 signal intensity follow from these mechanistic differences in a way we do not understand. Understanding this difference however seems beyond the scope of the paper.

- What are the Ca^{2+} dynamics after laser ablation of cell in the SAM? Now Ca^{2+} waves are described by prodding the meristem, and are connected to PIN polarization after cell ablation.

Response: A calcium wave response indicated by the GCaMP6f sensor upon cell ablation by a 2P-laser is shown in supplemental Movie 6. We have also shown PIN1 and microtubules responses to laser ablation upon Ca^{2+} inhibition (Figure 2). We checked and found that for GCaMP6f based wave response, it took 20 seconds for the wave to reach the periphery before the commencement of the withdrawal. This time frame is similar to R-GECO1 based wave response.

We would like to point out that a benefit of doing these experiments in two labs, and done differently, is scientific reproducibility - if the work is done independently in two labs, it is reproducible, especially if each lab does the experiment in their own way (pipette stimulation in the Meyerowitz lab, laser ablation in the Heisler, Zeiss laser scanning microscopy in the M lab, Leica resonance scanner in the H, R-GECO1 in M, GCaMP6f in H...).

- The last experiment shows that a LaCl_3 or BAPTA treatment after mechanical stimulation cannot prevent PIN relocalisation. However, it is unclear how long it takes for the inhibitors to penetrate the tissues sufficiently to impair Ca^{2+} signaling. This can be important information for correctly interpreting this experiment.

Response: We think the effect is as fast as we can move things around to treat and then look -10-15 minutes or shorter. As calcium wave inhibition responses and effect of PIN1 polarity in response to wounds showed that a 15-minute treatment of LaCl_3 and a 10-minute treatment of BAPTA were sufficient to inhibit the calcium wave. We can't move the plants and microscope around any faster. We clarified this detail in the main text (Page 13).

Reviewer #3 (Remarks to the Author):

In this manuscript Li et al explored the role of Ca²⁺ signaling in PIN1 polarization and subsequently its impact on auxin transport in the shoot apical meristem. Role of Ca²⁺ has been explored extensively in cell polarity from lower forms to higher forms of organisms. A large body of work devoted by the authors in the last decade to understand the role of PIN1 in auxin transport in shoot apex and how it gets polarized laid down a framework and many labs around the globe successfully used the resources and knowledge generated by them to characterize various mutants.

In this manuscript, the authors conducted a series of experiments to show that Ca²⁺ flux is generated within the shoot apical meristem cells in a periodic fashion evident from the live imaging experiments. This is generated perhaps via the release of Ca²⁺ ions in these cell types though it has no apparent correlation with any preferential PIN1 localization. By blocking the Ca²⁺ release the authors show that PIN1 polarization too get locked in the cells where it was initially expressed. This also has an impact on organogenesis. This observation sounds well and reasonable to expect if Ca²⁺ signaling plays such an important role. By these experiments authors conclude that the free Ca²⁺ ions availability is essential for normal functioning of shoot apical meristem because the PIN1 is also needed for normal phyllotaxy and auxin transport.

The authors also tested the possibility whether Ca²⁺ has any impact on microtubule orientation, which the authors standardized in the past in their laboratory. Based on the mechanical perturbation and needle injury experiments, the authors conclude that in the presence of LaCl₃, MT orientation does not get perturbed, however, the PIN1 polarization get affected. Suggesting that LaCl₃ blocks specifically PIN1 polarization and not act as a general inhibitor of ion transport. Hence PIN polarization and Ca²⁺ release is somehow linked and need to be dissected out to understand its precise role in cell polarity.

The authors could show with the available resources and tools that there is burst of Ca²⁺ in meristem, and does indeed influence the PIN1 polarization. The

findings presented in this study are novel, and would certainly help the wider community to appreciate the role of Ca²⁺ in meristematic cells.

However, the authors need to address following things.

Major comments

Despite the fact that LaCl₃ blocks calcium release, how specific it is and how to rule out the possibility that other channels related to Ca²⁺ signaling are not getting affected by this treatment. This aspect needs to be addressed in this article.

Response: We agree that pharmacological tests always come up with non-specific issues that are hard to avoid, this is the main reason we chose two different Ca²⁺ related chemicals to test our hypothesis. These two treatments work through entirely different mechanisms. Each of LaCl₃ and BAPTA may have non-specific effects, but they are not expected both to have the same non-specific effects, so that the common response to both should be the calcium response. And the experiment of wounding prior to LaCl₃ or BAPTA where PIN1 re-oriented in the presence of the inhibitors demonstrates that it is not due to an artefact that occurs after the initial Ca²⁺ signals (Fig. 4). This response was also given to reviewer 2, above, who had the same concern.

There are reports where LaCl₃ was also shown to block K⁺. The authors only focused on the PIN1 and strongly argue without testing other PIN transporter proteins.

Response: PIN1 is the main auxin efflux carrier in the SAM exhibiting dynamic patterns of expression and polarity and its loss of function is sufficient to disrupt phyllotaxis.

And, we get the same effect with BAPTA, which should not affect potassium.

It is evident from the LaCl₃ treatment that treated plants do not show a PIN-like shoot phenotype.

Response: We would like to point out that this short-term monitoring after drug treatment (12 hours) does not provide enough time for the shoot meristem to grow as a pin. Unfortunately, chemical inhibition of Ca^{2+} signals for significantly longer periods severely effects plant viability.

This study is still inconclusive from this reviewer's perspective. It explored the role of Ca^{2+} signaling in cell polarity but there is no evidence yet that suggest that Ca^{2+} get released in cell type specific manner

Response: Our hypothesis in general is that mechanical stress controls PIN1 and that Ca^{2+} increases are only required but are not sufficient for repolarization. We do not report or propose that Ca^{2+} is released in a cell type specific manner. In fact, we see responses all over the meristem. We clarify our conclusion in the main text abstract (page 2).

Minor comments

Why the authors choose to do the cortical microtubule orientation experiment test using 5mM LaCl_3 , whereas their other observation recorded at 3mM, if there are any specific reasons for this then it need to be addressed in the text.

Response: We double checked our text, all the observations were performed using 5mM LaCl_3 not 3mM. 3mM CaCl_2 was used during R-GECO1 or PIN1 reporter signal recovery after BAPTA treatment. And we revised our legends and text to provide greater detail about how each experiment was done including concentration and timing to avoid misunderstanding.

REVIEWERS' COMMENTS:

Reviewer #2 (Remarks to the Author):

The authors did a good effort to experimentally deal with most of the raised issues and used the results to improve their manuscript.

Minor remaining issues:

-Ref 12: 'Fuente' should probably be 'de la Fuente'

-I did not find a reference to a Figure showing "Partial inhibition of the relocation response could also be observed with a pretreatment of 1mM LaCl₃, but 5mM LaCl₃ completely prevented PIN1 repolarization". Maybe I missed it...

- the authors show the effect of the inhibitors on the calcium response, but it is not clear if this also correlates to the effects on PIN polarity.

- A simple google search led me to another paper concerning the link between calcium and PIN polarity, eg. Zhang et al., 2011 Dev Cell. I believe this could be relevant for making the discussion more general.

- That an early calcium signal regulates long term PIN polarity is striking, suggesting of a memory. This reminded me of an older paper where shoots responded at room temperature to a gravistimulus given under cold conditions (Perera et al 2006-Plant Phys). Do the authors think there could be a link?

Reviewer #3 (Remarks to the Author):

The importance of Ca²⁺ signaling in primordia formation specifically in the context of PIN1 polarization was not known till to date. In this manuscript, Li et al first time demonstrated evidence of Ca²⁺ release in the form of a burst using GCaMP6f and R-GECO1 sensor. Though the release of Ca²⁺ is random. However, similar observations were made in the past using the GCaMP6 in Zebrafish embryos too. It seems that the tools used by animal researcher can be applied successfully to unravel the role of Ca²⁺ in plants.

In the revised manuscript the authors addressed most of the concerns raised by this reviewer, therefore, I recommend it for publication in Nature Communication.

REVIEWERS' COMMENTS:

Reviewer #2 (Remarks to the Author):

The authors did a good effort to experimentally deal with most of the raised issues and used the results to improve their manuscript.

Minor remaining issues:

-Ref 12: 'Fuente' should probably be 'de la Fuente'

Thanks! We changed it in the main text.

-I did not find a reference to a Figure showing "Partial inhibition of the relocalization response could also be observed with a pretreatment of 1mM LaCl₃, but 5mM LaCl₃ completely prevented PIN1 repolarization". Maybe I missed it...

We added the images of PIN1-GFP with a pretreatment of 1mM LaCl₃ as Supplementary Figure 3a-e.

- the authors show the effect of the inhibitors on the calcium response, but it is not clear if this also correlates to the effects on PIN polarity.

In this study, we used two different treatments that inhibit the Ca²⁺ response by different mechanisms – LaCl₃ blocks plasma membrane-localized Ca²⁺ channels and BAPTA chelates apoplasmic Ca²⁺. Their application to SAMs prevents PIN1 protein relocalization after mechanical perturbation.

- A simple google search led me to another paper concerning the link between calcium and PIN polarity, eg. Zhang et al., 2011 Dev Cell. I believe this could be relevant for making the discussion more general.

Thanks to the reviewer pointing out the reference Zhang et al., we added it to the main text discussion section.

- That an early calcium signal regulates long term PIN polarity is striking, suggesting of a memory. This reminded me of an older paper where shoots responded at room temperature to a gravistimulus given under cold conditions (Perera et al 2006-Plant Phys). Do the authors think there could be a link?

This is an interesting point. We did not try mechanical perturbation under cold conditions to observe the Ca²⁺ response and PIN1 polarity and compare these reactions with room temperature controls, so it is not clear if the mechanism of Ca²⁺ signal regulation on PIN1 polarity is the same as the shoot gravity response.

Reviewer #3 (Remarks to the Author):

The importance of Ca²⁺ signaling in primordia formation specifically in the context of PIN1 polarization was not known till to date. In this manuscript, Li et al first time demonstrated evidence of Ca²⁺ release in the form of a burst using GCaMP6f and R-GECO1 sensor. Though the release of Ca²⁺ is random. However, similar observations were made in the past using the GCaMP6 in Zebrafish embryos too. It seems that the tools used by animal researcher can be applied successfully to unravel the role of Ca²⁺ in plants.

In the revised manuscript the authors addressed most of the concerns raised by this reviewer, therefore, I recommend it for publication in Nature Communication.